# The Diagnosis and Management of Autism Spectrum Disorder (ASD) in Adult Females in the Presence or Absence of an Intellectual Disability

**DOI:** 10.3390/ijerph19031315

**Published:** 2022-01-25

**Authors:** Tanzil Rujeedawa, Shahid H. Zaman

**Affiliations:** Cambridge Intellectual & Developmental Disabilities Research Group, Department of Psychiatry, University of Cambridge, Cambridge CB2 8AH, UK; mtr38@cam.ac.uk

**Keywords:** autism, female, intellectual disability

## Abstract

We review the reasons for the greater male predominance in the diagnosis of autism spectrum disorder in the non-intellectual disabled population and compare it to autism diagnosed in intellectually disabled individuals. Accurate and timely diagnosis is important, as it reduces health inequalities. Females often present later for the diagnosis. The differences are in core features, such as in social reciprocal interaction through ‘camouflaging’ and restricted repetitive behaviours, that are less noticeable in females and are potentially explained by the biological differences (female protective effect theory) and/or differences in presentation between the two sexes (female autism phenotype theory). Females more often present with internalising co-occurring conditions than males. We review these theories, highlighting the key differences and the impact of a diagnosis on females. We review methods to potentially improve diagnosis in females along with current and future management strategies.

## 1. Introduction

Autism Spectrum Disorder (ASD) is a neurodevelopmental condition that can present as a spectrum of symptoms, with different autistic individuals presenting differently. This condition is strongly genetic, with a heritability of around 80 to 90% [1]. In people with an intellectual disability (ID), ASD or autistic characteristics are more common and often associated with certain syndromes such as Fragile X, Down syndrome, Williams syndrome, Prader–Willi syndrome, Angelman syndrome, Rett syndrome, Noonan syndrome, and more [2]. It is important to point out that autism, besides being associated with several challenges, is also associated with certain positive traits such as truthfulness, honesty, integrity, high standards, and attention to details [3].

Throughout this paper, the term ‘autism’ will be used to represent all the conditions on the autism spectrum. In line with the preferences from a majority within the autism community, this paper will be using identity-first language (‘autistic person’) instead of person-first language (‘person with autism’) [4].

Though traditionally diagnosed in childhood, when differences from neurotypical peers become apparent, lately, there has been a surge in adult diagnosis due, in part, to the broadening of diagnostic criteria [5]. Due to the absence of any reliable biomarkers, autism is diagnosed behaviourally with diagnosis requiring the core characteristics: difficulties in reciprocal social communication and interactions, and ‘Restricted, Repetitive Behaviour’ (RRB), i.e., the presence of restricted interests and repetitive behaviours that lead to cognitive inflexibility and preference for routines. It is important to note the difficulty of diagnosis: the degree of impairment is not only dependent on the individual alone but also the surrounding environment. Symptoms are tamer in supportive settings, but they are more apparent in stressful situations [1]. This needs to be accounted for during diagnosis.

As diagnostic criteria broadened, the number of people considered as autistic has increased. For example, about 1 in 69 children in the US [6] and 1 in 59 in the UK [7] are affected by autism. Notably, such prevalence data comes, mainly, from well-developed western countries, as prevalence in low-income countries seems to be lower [8] either due to lack of proper diagnosis [9,10] or due to lifestyle factors.

The prevalence of autism in males and females is different, with higher prevalence in males. The estimated male to female ratio varies depending on the study and ranges from 4:1 [11] to 3:1 [12] to 2:1 [13]. Context is very important to consider; whole population screening leads to a lower ratio than when only considering clinically diagnosed individuals [12]. The ratio is also affected by presence of intellectual disability, with it reaching 2:1 when considering individuals with intellectual disabilities [14]. This could potentially be explained by reduced camouflaging in ID (see below). Besides the difference in prevalence, there is also a difference in age of diagnosis between the sexes, with females being generally diagnosed later [15,16,17]. The median age of diagnosis in men is 22 compared to 26 in women [18]. 

## 2. Methods

A structured review was undertaken, following the guidelines provided by Prisma Protocols [19]. First, a search in PubMed was carried out of the literature published in the last five years relating to autism in females, using the MeSH terms: (“Autism”[MeSH Terms] AND “Female”[MeSH Terms]). From the 1878 results, 269 were assessed for eligibility based on their relevance from their title and abstract, and from those, 54 were selected after a full-text review. A further search was conducted from literature published in the last 10 years using the MeSH terms: (“Autism”[MeSH Terms] AND “Intellectual disability”[MeSH Terms] AND “Female”[MeSH Terms]). From the 410 results, 200 were assessed for eligibility based on their relevance from their title and abstract, and from those, 16 were selected after a full-text review. The exclusion criteria applied to select the articles from their abstracts were the following: (a) *n* = 1, (b) only the title or abstract available, (c) opinion articles, clinical trials, novel test descriptions, or protocol revisions, (d) no specific content about autism in females, and (e) no specific reference to the impact of intellectual disability in autism. From the selected articles, we have finally included those that did not fulfil any of the previous exclusion criteria nor, following a full-text review, contain any new information. Finally, reference lists of selected papers were also searched for potentially relevant studies. The search methodology is summarised in Figure 1.

## 3. Explaining the Difference between the Sexes

We refer the reader to a review by Driver and Chester [20], and references therein, who summarise many of the clinical issues regarding autism in females. We will highlight some of these aspects here but with a focus on possible reasons for the differences and on autism in females with a non-syndromic intellectual disability.

There are two main theories that explain the sex difference observed in autism: notably, the Female Protective Effect (FPE) theory and the Female Autism Profile (FAP) theory. The two are not mutually exclusive and will be explored.

### 3.1. Female Protective Effect (FPE) Theory

The FPE theory argues that females are inherently protected from developing autism, requiring greater environmental and/or genetic risk to develop and manifest the same degree of autism as males [21]. This is supported by the fact that males, in general, have less spontaneous mutations associated with autism compared to females [22,23]. Autistic females also carry a larger size of rare copy number variations, containing genes that are expressed in the early development of the striatum [24].

The extreme male brain theory argues that androgens and other hormones prevalent in males are linked to autism, and characteristics associated with autism, such as impaired cognitive empathy, are masculine [25]. Autistic individuals, therefore, have an ‘extreme male’ presentation. It follows from this that females who have lower levels of these hormones are protected from the development of these male characteristics associated with autism [26,27]. Evidence supporting this theory is mixed, with some studies having found a link between androgen levels [28,29] and autistic characteristic in females, while others suggest otherwise [30]. Another theory proposed that the X chromosome has genes that protect against autism [31] but so far, no such gene has been found. Two peptide neurotransmitters, potentially playing a role in explaining the difference between male and female autism, are oxytocin (OXT) and arginine vasopressin (AVP), which have sexually dimorphic effects on brain activity, with stronger effects of OXT in females and stronger effects of AVP in males [32]. Both have been shown to have an effect on social behaviour [32,33,34].

It is important to note that the FPE theory has some limitations. If, as suggested, autistic females need a larger genetic load to develop autism, relatives of autistic females should be more likely to have autism compared to relatives of autistic males. Evidence around this is mixed, with some studies validating it [35,36], while others contradict it [37]. In addition, many of the studies cited assume that current diagnostic criteria are valid and that the autistic females participating have met the diagnostic criteria. However, this may not necessarily be true, as evidenced by the fact that the male to female ratio changes depending on the population used, as discussed above.

### 3.2. Female Autism Profile (FAP) Theory

The FAP theory proposes that women develop autism at a higher rate than currently estimated but that diagnostic criteria and methods are unable to properly detect autistic females [38,39]. Females tend to require additional difficulties, compared to males with similar levels of autistic characteristics, in order to receive a diagnosis [39,40,41,42]. This sex disparity is particularly prevalent in females without an intellectual disability [38]. The reason behind the inefficiency of the current diagnostic methods in identifying autistic females is that current criteria are based on pre-established conceptions of what autistic characteristics are. These conceptions are mainly based on autistic male populations [16,43,44]. Females present differently from males and, as such, may be missed under current criteria [45]. The female autism phenotype has the same core features of autism (impaired social communication abilities and RRB), though these may present differently from males [16,46,47].

Social impairments tend to be fewer in females, who, in general, also tend to display higher social motivation [48,49]. This difference in social abilities makes it harder to diagnose autistic females [50]. Higher social skills compared to autistic males, however, does not imply the absence of social challenges. Autistic females may be physically close to friends whilst not being accepted and suffering from neglect and exclusion [50]. In addition, autistic females seem to be less likely to maintain long term relationships compared to autistic males [49], potentially because autistic females are less apt at resolving social conflicts [51].

RRBs also present differently in the two sexes. Females seem to have lower levels of such behaviours [40,52,53] and interests, and RRBs seem to be less predictive of autism diagnosis in females compared to males [50]. There is an opposing theory that argues that instead of being lower, autistic females display RRBs in different, non-atypical areas compared to males [54,55]. Supporting this, studies have shown that autistic males’ interest seem to be on mechanical subjects [56,57], while autistic females’ interests seem to be on more relational subjects [56,57,58]. This type of interest, though the intensity may be atypical, is considered more socially appropriate compared to the interests of autistic males [59]. As a result of the either lower levels of RRBs, more socially acceptable RRBs, or both, autistic females are less likely to be identified [60].

Camouflaging refers to using strategies, whether consciously or unconsciously, to hide autistic characteristics within social settings [61]. These strategies include mimicking facial expressions, forcing eye contact, using scripted responses, and suppressing actions such as hand flapping [62,63,64]. Compensation, which can be considered as a subset of camouflaging, is a phenomenon whereby alternative cognitive strategies are employed to deal with the difficulties of autism, such as using executive function strategies to overcome theory of mind difficulties [65]. Camouflaging is generally associated with higher executive functions [66]. It would follow from this that autistic individuals with intellectual disabilities are less likely to camouflage their autistic traits, though no study validating this has been found. Camouflaging, being more prevalent in females, is a reason behind the difficulty of diagnosing female autism [67]. A study showed that, from a distance, autistic girls behaved similar to neurotypical peers, likely due to camouflaging, and it is only when properly looked into that the challenges of the autistic girls were identified [68]. Camouflaging does not, however, purely produce positive outcomes, with it being a taxing process that can lead to high levels of anxiety and stress [47]. It is also linked to mental health issues and is a known risk marker for suicidality [69,70]. Camouflaging can lead to a differential presentation across different social settings, as in less stressful situations, the autistic individual may not feel the need to camouflage [60]. This greater ability to camouflage or compensate autistic difficulties has been suggested to be due to female ability through genetics [16,39] thereby linking the FAP and FPE theories.

## 4. Implications of ASD Being Perceived as a ‘Male Disorder’

Due to the male-centric diagnostic criteria, higher prevalence of autism in males, and lack of information about female autism [71], autism has been perceived as a male disorder, and this is a reason for misdiagnosis of autism in females [72]. Cementing this is the fact that boys are 10 times more likely to be referred for diagnosis compared to girls [50]. This notion of autism as a male disorder can lead to a self-reinforcing cycle, whereby it leads to reduced identification of female autism. There seems to be more parental concern for ASD diagnosis for boys than for girls [50]. In addition, the concern of parents of girls tends to be met with scepticism by society [50]. There is also clinician bias whereby clinicians are more likely to exclude a diagnosis of autism for girls [15]. The notion of autism as a male-disorder is detrimental to the diagnosis of autistic females and should be combatted.

## 5. Associated Conditions

Autism is associated with several co-morbidities. For instance, autistic individuals are more likely to suffer from psychiatric disorders [73]. Between 35 and 42% of autistic people have an intellectual disability [74]. Co-morbidities associated with autism include anxiety, depression, bipolar affective disorder, obsessive compulsive disorder (OCD), attention deficit hyperactivity disorder (ADHD), psychosis, seizures, weight problems, hypertension, and many others [75]. These co-morbidities can either diagnostically overshadow autism, leading to missing a diagnosis of autism, or the characteristics of the co-morbidity are ascribed to autism, leading to other issues being missed. It is important to note that females are more likely to be diagnosed with these associated co-morbidities instead of autism compared to males [76]. While many of the co-morbidities do not discriminate between the sexes, some do, and these will be the focus of this section. 

Internalising disorders refers to the inward expression of emotional difficulties and can manifest as anxiety, depression, eating disorders, among others [77]. On the other hand, externalizing disorders refer to the outward expression of emotional difficulties and can present as behavioural problems and inattention. Autistic females are more likely to suffer from an internalising disorder compared to males [78,79,80], whilst males are more prone to externalizing disorders [49,81]. Internalising disorders being less apparent than externalizing ones is a reason behind the common missed diagnosis of female autism and the perception of autism as a male disorder [60].

Eating disorders are associated with autism, and particularly so in females [64,82], and they may be, in part, due to sensory integration problems in autism. There are two types of eating disorders associated with autism: avoidant/restrictive food intake disorder (ARFID) and anorexia nervosa (AN) [1]. In fact, many women have been identified as autistic following diagnosis of eating disorders [83]. AN and autism have many parallels, with both groups displaying aloofness, rigidity, RRBs, and social impairment [84,85]. Females with anorexia tend to have elevated autistic traits [85,86], thereby making it hard to distinguish whether the displayed phenotypes are part of AN or there is a co-occurrence of autism. One important distinguishing factor between the two conditions is the quality of social response: difficulties in social reciprocity is prevalent in ASD, while those suffering from AN show little difficulty in understanding social prompts [85,87].

Ehlers–Danlos syndrome (EDS), or hypermobility spectrum disorder (HSD), is another disorder commonly associated with female autism [88,89], often co-occurring within the same families [90]. Due to generally being treated by different disciplines, this co-morbidity often goes unnoticed [91]. It is important to note that, whilst these conditions may appear dissimilar, they, in fact, share several features such as coordination problems, sensory issues, autonomic dysregulation, immune dysregulation, and neurobehavioral, psychiatric, and neurological features [89].

Gender dysphoria and gender identity questioning are other conditions associated with autism [92]. Between 7.8 and 26% of referrals to gender clinics also have a diagnosis of autism [93,94,95]. This association between gender variance and autism is particularly prominent in females [64,96]. The extreme male brain theory provides a potential biological explanation [25].

## 6. Diagnosis in Women

Diagnosis of autism requires the clinical judgement of whether the core characteristics are met and whether their combined intensity is enough to cause a disability whilst ensuring not to confound autism with any other condition [1]. Helping in diagnosis is a series of screening questionnaires and interview frameworks. However, these diagnostic methods may not be suitable in identifying autistic females. For instance, assessments such as Autism Diagnostic Observation Schedule (ADOS) are more likely to miss girls [97]. This section will look at potential ways to diagnose autism in females. In addition, healthcare professionals are less confident in diagnosing autism in females [98].

During interviews, it is important to note sex differences in language. For instance, women tend to be slower to respond [99]. In addition, when examining social relationships, it is important to look deeper. Autistic individuals reporting friendships does not necessarily mean that those relationships are reciprocal [99], and a deeper dive, getting the patients to describe their feelings is necessary. Many autistic females describe feelings of loneliness despite reporting friendships [99].

Screeners looking at the female autistic presentation have been developed for children and adolescents, and when used in conjunction with other diagnostic tools, they can help better identify female autism [100]. Given that camouflaging is part of the female autism phenotype, assessing camouflaging is important in diagnosing women [101]. There are two main methods to measure camouflaging: the discrepancy approach and the observation approach.

The discrepancy approach aims to look at the difference between the internal autistic status of an individual and their external presentation [65]. A problem with this approach is that the only way to measure internal autistic status is by using external proxies. In addition, this method ignores unsuccessful camouflaging attempts. The observational approach seeks to either measure specific behaviours and experiences that are associated with camouflaging or asking the autistic individuals about camouflaging [102]. The self-report method eliminates biases from the clinicians and researchers.

The Camouflaging Autistic Traits Questionnaire (CAT-Q) seeks to measure camouflaging behaviours in adults using self-report measures [102]. The questionnaire has proven to be successful, and since there is no requirement for an official autistic diagnosis, it escapes the male centric biases of current diagnostic criteria [102]. As the CAT-Q relies on self-reflections, this method may not be suitable to individuals with intellectual disabilities [102]. Importantly, results from different studies using CAT-Q have been conflicting, with some studies arguing that camouflaging is not more prevalent in women [103] and others arguing it is [104]. This questions whether camouflaging is a ubiquitous feature of female autism. Whilst promising, the CAT-Q requires more research before becoming a clinical tool [102].

Female autism is often missed with current diagnostic tools: to solve this problem, many have investigated ways of identifying autistic females, such as by looking at camouflaging, which is a common reason behind missed diagnosis. However, so far, none has been approved for clinical use. Nonetheless, clinicians can help in better diagnosing females by dismissing the notion of autism as a male disorder and not dismissing concerns of parents. A clinician also needs to make sure that behind the mental illness diagnosis being made of a patient, there may be autism [20].

### Impact of Diagnosis in Women

Diagnosis of autism can have an immense impact on an individual. This section will focus mainly on the impact of autism diagnosis on females rather than the impact on both sexes, in general.

Autistic individuals with a normal intelligence are often labelled as high functioning [99], and this can lead to missed diagnosis and the notion that they are in no need of support. This external high functioning presentation can lead to discrediting of the struggles of autistic individuals, especially females using camouflaging, by therapists [99]. The high functioning outer presentation means that autistic females appear neurotypical to society and clinicians, leading to missed diagnosis. As such, there is an expectation for proper and socially aware behaviours [99]. Girls with undiagnosed autism are often labelled as lazy or rude and suffer from bullying [99]. They may be criticized by parents and teachers due to not abiding by neurotypical behaviours despite being seemingly neurotypical [99].

Sexual education is another important issue that undiagnosed autistic females may struggle with. Due to limited formal sex education, peer to peer education about norms and safety is important and such informal education may be missed by autistic adolescents [99]. This may put autistic females at risk of sexual abuse, and indeed, there is a high incidence of abuse in this demographic [70]. Consequently, autistic women should be provided with the necessary information and education to allow them to navigate their sexual and reproductive health [105].

Undiagnosed autistic females have been misunderstood and negatively labelled, and a proper diagnosis can be immensely helpful. Even high functioning autistic individuals need support [99]. Diagnosis would allow professionals to provide needed support, validate needs, and help address issues. Furthermore, diagnosis facilitates support such as disability benefits and job inclusion [18]. It was also found that diagnosis helped foster a positive sense of identity in women [70]. Moreover, a diagnosis can provide a rationale to the behaviours and combat criticisms from society, thereby alleviating blame [99]. Diagnosis has also been shown to provide an explanation to past experiences, allowing the individual to make sense of them [99].

After a diagnosis is made, women may initially feel distress [106], as diagnosis signifies the end of a dream for normality [99]. Therefore, it is important to highlight potential strengths.

## 7. Management

### 7.1. Current Management

This section will cover the management and treatments of autistic individuals, in general, which are limited. There are no specific treatments for autism for females, but being aware of the differences in presentation and diagnosing and managing the co-morbidities is essential. It is important to distinguish between the treatment of autism versus the treatment of associated co-morbidities.

Management of autism normally involves strategies to reduce the impact of the autistic traits on daily functioning, such as providing the training and support needed to function independently [107]. This rehabilitation approach aims to increase physical comfort, reduce anxiety, and improve the environment, such as by making it more predictable and reducing sensory stimuli overload, which are helpful [1]. Autistic individuals, in particular girls, should be educated in sexuality and safety due to the ineffectiveness of peer to peer learning in this population [99]. There is also a need for wider societal acceptance of neurodiverse behaviours and for that, education is important.

Psychosocial and behavioural therapies are the recommended therapies for autism [108,109]. It is important to note that RRBs may be coping strategies in autism, and as such, suppressing these behaviours may be deleterious [18]. Autistic individuals suffer from bullying and victimization, and as such, trauma informed care is important [1]. It is necessary to provide a safe setting whereby the experiences of autistic individuals are validated [1]. Consciousness raising is believed to be beneficial for autistic females, as it allows them to untangle social expectations from their own desires [105].

Whilst there is no pharmacological treatment of autism itself, such treatments are used for the co-morbidities. It is important to note that tangible treatments can have positive placebo effects on autistic individuals, either directly or indirectly [110]. Additionally, autism (and more so in ID) seems to be a marker of unpredictable adverse responses to psychotropic drugs, and as such, drugs should be cautiously introduced at low doses [1]. There is also a noted difference in the treatment offered to females compared to males: females are more likely to be offered sedatives, while males are more likely to be prescribed a combination of medication [111]. In addition, when providing pharmacological treatment, special attention should be provided to sex, due to the risk of female specific side effects [105]. For instance, aripiprazole is one of the few atypical antipsychotic medications, typically used to treat irritability in autistic children [112], that does not cause hyperprolactinemia and, subsequently, gynaecomastia and galactorrhea. Additionally, certain medications used in co-morbid ASD such as modafinil for ADHD and the anticonvulsant carbamazepine, can reduce the efficacy of medications used for contraception. These risks should always be kept in mind and communicated to patients.

### 7.2. Future Management

This section will explore the future of autism management.

Consuming cannabis is believed to improve communication skills [113]. Cannabidiol, a component of cannabis, is anxiolytic and neuroprotective [114]. This has led to the idea of using cannabinoids as a treatment for autism [114].

There is some evidence supporting the use of glutamate neurotransmitter-modulating agents in autism [115]. Memantine, which has been used as an adjunct therapy for ADHD and OCD is one such example [116]. Use of Memantine in children and adolescents has been positive, leading to reduced irritability and improved social interactions [117].

The autonomic nervous system seems to play a role in social behaviour and emotion regulation. Vagus nerve stimulation, a neuromodulation technique that stimulates autonomic pathways, has been proposed as a treatment for autism [118]. There is some evidence, though not conclusive, that vagus nerve stimulation may improve behaviour in autism [118].

Although autistic individuals are more likely to be diagnosed with gastrointestinal disorders such as constipation [119], there is evidence of the influence of the gut microbiota on autism [120]. Work on gut microbiota, as a treatment modality, are still under development [121].

Precision medicine is another potential avenue for autism treatment. Rare monogenic forms of autism have allowed the identification of pathways as potential therapeutic targets [122]. Despite this rarity, common pathways seem to be involved in autism [123]. Treatments identified in this way, therefore, could be applied to broad patient groups, and this implies the need to sub-classify autism based on the genes and pathways involved.

It is important to point out that more research is needed for the treatments discussed in this section, and yet, these are only in early stages.

## 8. Conclusions

Autism appears to be less prevalent in females compared to males, with the male to female ratio varying from 2:1 to 4:1, depending on the study and population (including the ID population). The reason behind this difference is explained by two non-mutually exclusive theories. The female protective effect theory argues that women have an inherent protection against autism, while the female autism phenotype theory argues that current diagnostic criteria, being biased towards males, can miss autistic females who present differently. Autistic females seem to face less difficulties in social interactions, and their RRBs seem to either be lower or more socially acceptable. In addition, females tend to exhibit more camouflaging behaviours, which is associated with depression and suicidality. Table 1 provides a summary of the differences between autistic males and females. All of this, together with the societal perception of autism as a male disorder, makes it easier to miss a diagnosis of autism in females. In people with ID and associated cognitive impairment (such as executive function) may make it harder for them to use camouflaging, thus explaining the lesser male to female disparity. Diagnosis of autism in women needs to take these differences into account, and tools looking at the female presentation of autism have been created. More research, however, is necessary for these tools to be used in the clinic. Diagnosis of autism in females is important, as it can allow them to feel validated and makes it easier to provide support. Autism is associated with several co-morbidities and those common in autistic females include internalising disorders, eating disorders, and hypermobility spectrum disorders. The treatment and management of these co-morbidities need to be distinguished from that of autism itself. The management and treatment of autism primarily seeks to lessen the functional impact of the autistic characteristics rather than eliminating autism from the individual. There are interesting therapeutic avenues for the future, ranging from the use of cannabinoids to gut microbiota to precision medicine, which will hopefully improve the lives of autistic individuals.

## Figures and Tables

**Figure 1 ijerph-19-01315-f001:**
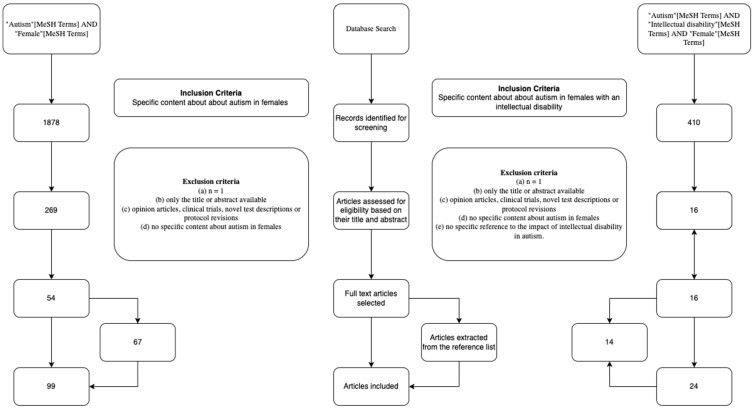
Search methodology undertaken for this review.

**Table 1 ijerph-19-01315-t001:** Summary of the differences between the sexes.

Aspect or Feature	Male	Female
**Prevalence**	4-2	1
**Impaired Social Communication Abilities**	Present	Present, but generally females have higher social abilities
**Restricted Repetitive Behaviours**	Present	Present, but either lower or displayed in more sociably acceptable areas (i.e., more relational than mechanical)
**Camouflaging**	Present	More prevalent than males
**Associated Conditions**	Externalising co-occurring disorders (e.g., behavioural problems and inattention) more likely	Internalising co-occurring disorders (e.g., anxiety, depression and eating disorders) more likely

## Data Availability

Not applicable.

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
