# Peer review of "The Diagnosis and Management of Autism Spectrum Disorder (ASD) in Adult Females in the Presence or Absence of an Intellectual Disability"

_ijerph, 2022, doi:10.3390/ijerph19031315_

Round 1

Reviewer 1 Report

Thank you for asking me to read and comment on the review titled: "The Diagnosis and Management of Autism Spectrum Disorder(ASD) in Adult Females in the Presence or Absence of an Intellectual Disability". The study is timely and would be interesting for the journal audience.

I do have some suggestions:

  1. Edit the abstract to make the reader understand the bottom line of the findings
  2. Add a plot for the PRISMA 
  3. there are missing spaces before each [
  4. Section 7 is about treatment in general, but it seems not specific for the current Review.
  5. There is no reference to the studies on OXTR and AVP genes and their implication on ASD (and these two hormones tend to be gender-specific, so I was expecting to find some discussion about that in the text)
  6. Another relevant domain, relevant to ASD and etiology is the interaction of the GUT and Brain. I wonder if gender-specific effect in this domain might be described (if any)

Author Response

We thank you Rev 1 for your time and comments to help improve the piece.

  1. We have made changes to the abstract (copied below):

We review the reasons for the greater male predominance in the diagnosis of autism spectrum disorder in the non-intellectual disabled population and compare it to autism diagnosed in intellectually disabled individuals. Accurate and timely diagnosis is important as it reduces health inequalities. Females often present later for the diagnosis. The differences are in core features such as in, social reciprocal interaction through ‘camouflaging’ and restricted repetitive behaviours that are less noticeable in females are potentially explained by the biological differences (female protective effect theory) and/or differences in presentation between the two sexes (female autism phenotype theory). Females more often present with internalising co-occurring conditions than males.  We review these theories; highlight the key differences and the impact of a diagnosis on females. We review methods to potentially improve diagnosis in females along with current and future management strategies. 

  1. We had added a revised section on p2 of the updated manuscript.

  1. We have corrected this.

  1. We have added a comment that most of what we know regarding treatment is not sex-specific. However, there is a section specific to females regarding their understanding of relations and its impact on sexual vulnerability.

  1. Thank you for highlighting these important sex-related biological factors. We have added information regarding these peptides (please see line 104 and onwards).

  1. Thank you again for pointing out the omission of this evolving area. We have added a piece from line 360 and onwards.

SUMMARY OF ALL CHANGES MADE (lines refer to updated version):

  1. Abstract (as above)
  2. Flow diagram to indicated methodology on page 2
  3. AVP and OXT involvement in autism: from line 104:
  4. Clarification of sentence regarding social deficit manifestation in females: line 131
  5. Comment regarding management of autism specific to females: from line 303 to 315.
  6. Comment regarding future management strategies and precision medicine: from line 356 to 360.
  7. A summary table to highlight differences between males and females added.
  8. Any typos or grammatical errors have been corrected.

Reviewer 2 Report

Rujeedawa and Zaman proposed an interesting review article aimed at elucidating the main features of autism spectrum disorder in adult females to improve the diagnosis and management of this disease. Overall, the manuscript is interesting and well-structured, however, there are some issues to clarify:
1) Please better emphasize the coherence of the manuscript with the aims of IJERPH;
2) Please check the following sentence: “While social impairments are present in females, they tend to be fewer in females,...”. Is the first “female” correct?;
3) The authors have to provide the schematic workflow used for the selection of the studies;
4) According to the previous comment, the authors should present some clinical-pathological data derived from the analyzed studies in a tabular format.

Author Response

We appreciated your time and effort to review this. We welcome your comments and we have tried to address them as below.

1) The main management of this condition is through a rehabilitation approach, where environmental modifications, paying particular attention to social interactions and reducing the impact of sensory integration dysfunction allows the person to thrive more favourably. The available treatments thus far aim not to “change the autism”, being a neurodevelopment condition. So, we are of the opinion that this fits the remit of this special issue as it addresses the problem with the late diagnosis or late presentation and the lack of recognition of autism in females that leads to health inequalities.  I have added a sentence in the abstract to point this out.

2) Thank you for highlighting this confused sentence. We changed this to (line 131):

Social impairments tend to be fewer in females, who in general also tend to display higher social motivation [48, 49]. This difference in social abilities makes it harder to diagnose autistic females [50].

3) We had added a revised section on p2 of the updated manuscript.

4) This is a really helpful suggestion. We have added a table in the updated manuscript.

SUMMARY OF ALL CHANGES MADE (lines refer to updated version):

  1. Abstract (as above)
  2. Flow diagram to indicated methodology on page 2
  3. AVP and OXT involvement in autism: from line 104:
  4. Clarification of sentence regarding social deficit manifestation in females: line 131
  5. Comment regarding management of autism specific to females: from line 303 to 315.
  6. Comment regarding future management strategies and precision medicine: from line 356 to 360.
  7. A summary table to highlight differences between males and females added.
  8. Any typos or grammatical errors have been corrected.

Reviewer 3 Report

This review article used method of PRISMA protocol to find and review scientific literature about adult female Autism Spectrum Disorder in the presence or absence of an intellectual disability. The manuscript written is fluent and interesting; however, I suggest:

  1. The abstract is too short; readers cannot get enough information from abstract link to the article title.
  2. Method, lines 57-73, is too simple. It is lack of a PRISMA flow diagram, nor the checklist.

If they can improve the above two points, this review article could be more helpful to readers who are interested or in related fields.

Author Response

We thank you Rev3 for your time and comments.

1) We have improved the abstract:

We review the reasons for the greater male predominance in the diagnosis of autism spectrum disorder in the non-intellectual disabled population and compare it to autism diagnosed in intellectually disabled individuals. Accurate and timely diagnosis is important as it reduces health inequalities. Females often present later for the diagnosis. The differences are in core features such as in, social reciprocal interaction through ‘camouflaging’ and restricted repetitive behaviours that are less noticeable in females are potentially explained by the biological differences (female protective effect theory) and/or differences in presentation between the two sexes (female autism phenotype theory). Females more often present with internalising co-occurring conditions than males.  We review these theories; highlight the key differences and the impact of a diagnosis on females. We review methods to potentially improve diagnosis in females along with current and future management strategies. 

2) We had added a revised section on p2 of the updated manuscript.

SUMMARY OF ALL CHANGES MADE (lines refer to updated version):

  1. Abstract (as above)
  2. Flow diagram to indicated methodology on page 2
  3. AVP and OXT involvement in autism: from line 104:
  4. Clarification of sentence regarding social deficit manifestation in females: line 131
  5. Comment regarding management of autism specific to females: from line 303 to 315.
  6. Comment regarding future management strategies and precision medicine: from line 356 to 360.
  7. A summary table to highlight differences between males and females added.
  8. Any typos or grammatical errors have been corrected.

Round 2

Reviewer 3 Report

I have no more comments, since the authors revised the manuscript as reviewers' suggestions.